# Betavulgarin Isolated from Sugar Beet (*Beta vulgaris*) Suppresses Breast Cancer Stem Cells through Stat3 Signaling

**DOI:** 10.3390/molecules25132999

**Published:** 2020-06-30

**Authors:** Ren Liu, Hack Sun Choi, Xing Zhen, Su-Lim Kim, Ji-Hyang Kim, Yu-Chan Ko, Bong-Sik Yun, Dong-Sun Lee

**Affiliations:** 1Interdisciplinary Graduate Program in Advanced Convergence Technology and Science, Jeju National University, Jeju 63243, Korea; liuren0308@jejunu.ac.kr (R.L.); zhenxing0213@jejunu.ac.kr (X.Z.); ksl1101@naver.com (S.-L.K.); seogwi12@naver.com (J.-H.K.); uchan@jejunu.ac.kr (Y.-C.K.); 2Subtropical/Tropical Organism Gene Bank, Jeju National University, Jeju 63243, Korea; choix074@jejunu.ac.kr; 3Division of Biotechnology, College of Environmental and Bioresource Sciences, Jeonbuk National University, Gobong-ro 79, Iksan 54596, Korea; bsyun@jbnu.ac.kr; 4Practical Translational Research Center, Jeju National University, Jeju 63243, Korea; 5Faculty of Biotechnology, College of Applied Life Sciences, Jeju National University, SARI, Jeju 63243, Korea

**Keywords:** breast cancer stem cells (BCSCs), betavulgarin, mammospheres, Stat3, SOX2

## Abstract

Breast cancer is a major health problem that affects lives worldwide. Breast cancer stem cells (BCSCs) are small subpopulations of cells with capacities for drug resistance, self-renewal, recurrence, metastasis, and differentiation. Herein, powder extracts of beetroot were subjected to silica gel, gel filtration, thin layer chromatography (TLC), and preparatory high-pressure liquid chromatography (HPLC) for isolation of one compound, based on activity-guided purification using tumorsphere formation assays. The purified compound was identified as betavulgarin, using nuclear magnetic resonance spectroscopy and electrospray ionization (ESI) mass spectrometry. Betavulgarin suppressed the proliferation, migration, colony formation, and mammosphere formation of breast cancer cells and reduced the size of the CD44^+^/CD24^−^ subpopulation and the expression of the self-renewal-related genes, *C-Myc*, *Nanog*, and *Oct4*. This compound decreased the total level and phosphorylated nuclear level of signal transducer and activator of transcription 3 (Stat3) and reduced the mRNA and protein levels of sex determining region Y (SRY)-box 2 (SOX2), in mammospheres. These data suggest that betavulgarin inhibit the Stat3/Sox2 signaling pathway and induces BCSC death, indicating betavulgarin might be an anticancer agent against breast cancer cells and BCSCs.

## 1. Introduction

Recently, the study of the biological activity of red beetroot (*Beta vulgaris rubra*) has been growing. Beetroot is a healthy vegetable rich in anthocyanin, betacyanin, folic acid, phenolic compounds, ascorbic acid, flavonoids, vitamin C, and other biologically active components [1,2,3]. Specifically, beetroot contains betalains, which are known for their antioxidant [2,4,5], anti-inflammatory [5], anticancer [1,5,6], and chemopreventive bioactivities [7]. Moreover, beetroot, as a cost-effective strategy, is considered an effective juice supplement for therapeutic treatment of clinical pathologies related to oxidative stress and inflammation [3].

Breast cancer is a serious health problem diagnosed in women [8]. The treatment options for breast cancer include mastectomy or breast-conserving surgery, hormone therapy, chemotherapy, and radiotherapy. However, patients treated with conventional therapies suffer from breast cancer relapse and metastasis [9]. The existence of cancer stem cells (CSCs) was first identified by Bonnet and Dick [10]. BCSCs are self-renewing and contribute to tumor recurrence; breast cancer stem cells (BCSCs) were first isolated by Al-Hajj [11]. A subpopulation of breast cancer cells exhibit the surface phenotype CD44^+^CD24^−^. Subsequently, this subpopulation can be detected in circulating breast cancer cells from patients and is associated with breast cancer recurrence and distant metastasis [12]. It is essential to target BCSCs in cancer treatment. Stemness proteins and signaling pathways involved in BCSC maintenance include embryonic stemness transcription factors (octamer-binding transcription factor 4 (Oct4), sex determining region Y (SRY)-box 2 (SOX2), c-myelocytomatosis (c-Myc), and Chromosome-associated kinesin (KIF4A), the canonical and noncanonical wnt pathways, Notch and the hedgehog pathway [9].

The oncogenic transcription factor Signal transducer and activator of transcription 3 (Stat3) is associated with cancer progression, metastasis, chemoresistance, stem cell self-renewal and maintenance, autophagy, and immune evasion [13,14,15]. In triple-negative breast cancer (TNBC), Stat3 is constitutively activated and highly related to poor survival outcomes [16]. Many studies have also demonstrated that Stat3 phosphorylation and activation upregulate the expression of cMyc and Sox2, which promote the self-renewal of breast cancer cells [17,18,19,20].

In this study, we selected Jeju beetroot to target BCSCs, and the active component was isolated based on activity-guided fractionation. Betavulgarin, the isolated active component, inhibited BCSC formation. We demonstrate that betavulgarin suppresses the proliferation of breast cancer and BCSC formation through the regulation of Stat3/Sox2 signaling in BCSCs.

## 2. Results

### 2.1. Isolation of a BCSC Inhibitor from Beta vulgaris

To screen and purify BCSC inhibitors from *Beta vulgaris rubra*, a mammosphere formation assay using MDA-MB-231 cells was performed, and a BCSC inhibitor was purified using methanol extracts of *Beta vulgaris rubra* generated by ethyl acetate extraction, silica gel filtration, Sephadex LH-20 (GE Healthcare, Uppsala, Sweden) chromatography, preparatory thin-layer chromatography (TLC), and preparatory high-pressure liquid chromatography (HPLC) (Figure 1A). The purified compound suppressed BCSC formation (Figure 1B) and was analyzed using HPLC (Figure 1C). The molecule identified using nuclear magnetic resonance (NMR) data was determined to be betavulgarin (Figure 2).

### 2.2. Betavulgarin Inhibits Breast Cancer Cell Growth and Mammosphere Formation

To ascertain whether betavulgarin has an inhibitory effect on breast cancer growth, we assessed the inhibitory effect of betavulgarin at increased concentrations in MDA-MB-231 and MCF-7 cells. Betavulgarin had an antiproliferative effect on the MDA-MB-231 and MCF-7 cells at ≥ 100 µM and 50 µM, after 24 hrs of treatment (Figure 3A,B). To confirm whether betavulgarin can suppress mammosphere formation, it was added to mammospheres derived from MDA-MB-231 or MCF-7 cells. As shown in Figure 3C,D, betavulgarin decreased not only the sphere numbers of MDA-MB-231 and MCF-7 cells by 78% and 68%, respectively, but also the sizes of the mammospheres. In addition, betavulgarin inhibited migration and colony formation of MDA-MB-231 and MCF-7 cells (Figure 3E,F). We showed that betavulgarin inhibits mammosphere formation, migration, colony formation, and breast cancer growth.

### 2.3. Betavulgarin Decreases CD44^+^/CD24^−^-Expressing Cancer Cell Numbers

The phenotype indicative of BCSCs was CD44^+^/CD24^−^. The CD44^+^/CD24^−^ cell fraction of MDA-MB-231 cells was examined after betavulgarin treatment. Betavulgarin reduced the CD44^+^/CD24^−^ cell fraction of MDA-MB-231 cells from 90.4% to 57.2% (Figure 4).

### 2.4. Betavulgarin Inhibits the Nuclear Translocation of Stat3 in BCSCs

To examine the biochemical mechanism underlying the suppression of mammosphere formation by betavulgarin, we examined the total protein levels of Stat3, p-Stat3, and NF-κB p65. Our data showed that the levels of Stat3 and p-Stat3 were significantly decreased following betavulgarin treatment (Figure 5A). The levels of nuclear Stat3, p-Stat3, and p65 were determined, and these results showed that the nuclear Stat3 and p-Stat3 levels were significantly reduced by betavulgarin but those of p65 were not (Figure 5B). Furthermore, an immunofluorescence (IF) assay assessing pStat3 was performed in MDA-MB-231 cells, and the level of nuclear pStat3 in betavulgarin-treated cancer cells was lower than that in control cells (Figure 5C). Moreover, we examined the direct binding of a Stat3 binding probe to Stat3 proteins under betavulgarin treatment, using an electrophoretic mobility shift assay (EMSA) (Figure 5D). We examined nuclear Stat3-specific DNA binding using an Infrared Dye (IRDye)-labeled Stat3 probe that bound to Stat3 proteins under betavulgarin treatment. Our data showed that the amounts of nuclear Stat3 proteins bound to the IRDye-labeled Stat3 probe (indicated by arrow) were significantly decreased by betavulgarin treatment (Figure 5D, line 3). The specific binding of the Stat3 proteins/probe was confirmed using a self-competitor (Figure 5D, line 4) and a mutated Stat3 oligo (Figure 5D, line 5). Recently, it was reported that Stat3 protein binds to the promoter region of the SOX2 gene and increases SOX2 transcription. Stat3/SOX2 regulates the self-renewal of lung CSCs [17,18,19,20]. After betavulgarin treatment, we checked the Sox2 level because the Stat3 dimer activates the Sox2 gene. Our data showed that betavulgarin decreased the transcript and protein levels of Sox2 (Figure 5E). Our data showed that Stat3/Sox2 signaling was important in mammosphere formation.

### 2.5. Betavulgarin Inhibits the mRNA Levels of BCSC-Specific Marker Genes and Mammosphere Growth

To examine whether betavulgarin reduced the mRNA levels of BCSC marker genes, we determined the mRNA levels of these genes. Betavulgarin reduced the transcriptional levels of the BCSC marker genes (Figure 6A). To check whether betavulgarin decreased mammosphere growth, we cultured mammospheres with betavulgarin and counted the number of mammosphere cancer cells. Betavulgarin increased cell death and reduced mammosphere growth (Figure 6B).

## 3. Discussion

Red beetroot (*Beta vulgaris var. rubra L*.) contains many bioactive compounds, including anthocyanin, betacyanin, folic acid, phenolic compounds, ascorbic acid, flavonoids, vitamin C, and other biologically active components. The most important bioactive phytochemicals in red beetroot are betalains, a class of tyrosine-derived pigments obtained from betalamic acid, whose members are grouped into yellow betaxanthins and red betacyanins. Red dye E162 extract from beetroot is approved for use in the food industry by the European Food Safety Authority. Betalains have been demonstrated to have strong free radical scavenging, antioxidant [5,21,22], and anti-inflammatory activities [23,24]. In this report, we isolated a BCSC inhibitor, betavulgarin, based on activity-guided fractionation. Betavulgarin was reported to be a fungus infection response molecule and an antifungal molecule in beetroot [25]. For the first time, we report that betavulgarin inhibits BCSCs.

Breast cancer is the most frequent cancer among women [8]. Breast cancer is a systemic disease characterized by early tumor cell dissemination and displays a high degree of intratumor heterogeneity that is important for therapeutic resistance, recurrence, and tumor progression [26,27]. Recently, a BCSC model was proposed and has received increasing interest in the field. CSCs are characterized by the common features of stem cells, including static behaviors, self-renewal, and differentiation.

Achieving efficacious breast cancer treatment is challenging because of the existence of BCSCs. Numerous pathways and factors that could be targeted to inhibit BCSCs were identified. Our results showed that betavulgarin inhibits the proliferation of MDA-MB-231 and MCF-7 cells (Figure 3A,B) and the size and number of mammospheres derived from MDA-MB-231 or MCF-7 cells (Figure 3C,D). To address changes in the diverse biological properties of breast cancer cells under betavulgarin treatment, cell migration and colony formation were tested in the context of betavulgarin treatment. Our results showed that betavulgarin inhibits the migration and colony formation of human breast cancer cells (Figure 3E,F). Additionally, betavulgarin reduced the size of the CD44^+^/CD24^−^ subpopulation in breast cancer cells (Figure 4). It is known that BCSCs are substantially regulated by a multitude of signaling pathways and transcription factors (such as Notch, Hedgehog, Wnt pathways, NF-kB, and Stat3), and that targeting these pathways represents a potential therapeutic approach [9]. In this regard, we explored the role of betavulgarin in the inhibition of BCSCs. Interestingly, the expression levels of Stat3 and p-Stat3 were downregulated by betavulgarin, as was the nuclear localization of Stat3 (Figure 5). It was reported that natural products such as quercetin, apigenin, oroxylin A (flavones), butein (chalcone), piperlongumine, and caffeic acid (hydoxycinnamic acid) act as stat3 inhibitors [28]. Betavulgarin belongs to isoflavone and might be a small-molecule inhibitor of Stat3 because of a similar structure of flavone. The activation of several transcriptional factors related to embryonic stem cell growth and differentiation, such as sex determining region Y (SRY)-box 2 (SOX2), could explain the enhanced stemness of BCSCs, compared to that of non-BCSCs [9]. One key transcription factor regulating SOX2 expression is Stat3, which directly binds to the promoter of SOX2 [29]. Subsequently, after treatment with betavulgarin, the mRNA transcription and protein expression of SOX2 were assessed, and the results showed that betavulgarin inhibited SOX2 through Stat3 inhibition (Figure 5). Betavulgarin reduced the transcriptional levels of the C-Myc, Nanog, and Oct4 genes and decreased mammosphere growth (Figure 6). Our data suggest that betavulgarin, which targets Sox2/Stat3 signaling, might be used as an anti-cancer agent.

## 4. Materials and Methods

### 4.1. Chemical and Reagents

Silica gel 60 and TLC plates were purchased from Merck (Darmstadt, Germany), and Sephadex LH-20 was obtained from Pharmacia (Uppsala, Sweden). Cell viability was measured using the EZ-Cytox Cell Viability Assay Kit (DoGenBio, Seoul, Korea). Other compounds were obtained from Sigma-Aldrich (St. Louis, MO, USA).

### 4.2. Plant Material

A sample of beet was obtained from verified market sources (Seogwipo, Jeju, Korea). The beets were washed and freeze-dried, and the dried beet was ground. A voucher specimen (No. 2018_010) was deposited in the Department of Biomaterial, Jeju National University (Jeju-Si, Korea).

### 4.3. Isolation of a Mammosphere Formation Inhibitor form Beat

The ground samples of beet were extracted with methanol. The isolation method is summarized in Figure 1A. The beet powder was solubilized with 10 L of methanol. The methanol extracts were concentrated and mixed with equal volumes of water, and the methanol part was evaporated. The water-suspended part was extracted with equal volumes of ethyl acetate. The solubilized ethyl acetate-concentrated part was loaded onto a silica gel column (3 × 35 cm) and eluted with a solvent (chloroform–methanol, 10:1) (Appendix A). Five fractions were divided and assayed by evaluating mammosphere formation. The #2 fraction potentially suppressed mammosphere formation. The #2 fraction was loaded onto a Sephadex LH-20 open column (2.5 × 30 cm) and fractionated into four fractions (Appendix A). The four fractions were further fractionated and analyzed by evaluating the mammosphere formation. Part #4 showed inhibition of mammosphere formation. Part #4 was isolated using preparatory TLC (glass plate; 20 × 20 cm) and developed in a TLC chamber. Individual bands were separated, and each fraction was assayed by evaluating the mammosphere formation (Appendix A). The #1 fraction was loaded onto a Shimadzu HPLC instrument (Shimadzu, Tokyo, Japan). HPLC was performed with an ODS 10 × 250-mm column (flow rate; 2 mL/min). For elution, the acetonitrile proportion was initially set at 30%, increased to 60% at 20 min, and finally increased to 100% at 30 min (Appendix A).

### 4.4. Structure Analysis of the Purified Compound

The chemical structure of the compound was determined by ESI-mass spectrometry and NMR spectroscopy measurements. The molecular weight was estimated to be 312 by ESI-mass spectrometry, which showed a quasi-molecular ion peak at m/z 313.3 [M + H]^+^ in the positive mode (Appendix A). The ^1^H NMR spectrum measured in CDCl_3_ exhibited signals due to a hydroxyl proton at δ 9.02, and four aromatic methine protons at δ 7.32, 7.09, 7.07, and 6.93, which could be attributed to a 1,2-disubstituted benzene ring; two aromatic singlet methines at δ 7.90 and 6.70; a dioxymethylene at δ 6.10; and a methoxy group at δ 4.11. In the ^13^C NMR spectrum, the 17 carbon peaks included a carbonyl carbon at δ 178.7; five oxygenated sp^2^ quaternary carbons at δ 156.7, 154.7, 153.8, 141.4, and 135.8; one oxygenated sp^2^ methine carbon at δ 153.4; five sp^2^ methine carbons at δ 130.4, 130.0, 120.6, 119.4, and 92.9; three sp^2^ quaternary carbons at δ 125.7, 120.8, and 112.8; one dioxymethylene carbon at δ 102.4; and one methoxy carbon at δ 61.3 (Appendix A). All proton-bearing carbons were assigned by the HMQC spectrum, and the ^1^H-^1^H COSY spectrum revealed a partial structure of 1,2-disubstituted benzene (Appendix A). Further structural elucidation was performed with the aid of the HMBC spectrum, which showed long-range correlations from the methine proton at δ 7.90 to the carbons at δ 178.7, 154.7, 125.7, and 120.8; from the methine proton at δ 7.09 to the carbons at δ 156.7 and 125.7; from the methine proton at δ 6.70 to the carbons at δ 154.7, 153.8, 135.8, and 112.8; and from the dioxymethylene protons to the carbons at δ 153.8 and 135.8. Finally, a methyl proton showed a long-range correlation to the carbon at δ 141.4 (Appendix A). Therefore, the structure of the isolated compound was identified as that of betavulgarin (Figure 2).

### 4.5. Culture of Human Breast Cancer Cells and Mammospheres

MCF-7 (ATCC^®^ HTB-22^TM^) and MDA-MB-231 (ATCC^®^ HTB-26^TM^) breast cancer cell lines were purchased from the American Type Culture Collection (Rockville, MD, USA) and incubated in Dulbecco’s modified Eagle’s medium (DMEM) supplemented with 10% (*V*/*V*) fetal bovine serum (Gibco, ThermoFisher, CA, USA) and 1% penicillin/streptomycin (Gibco, ThermoFisher, CA, USA) in a 5% CO_2_ incubator. Breast cancer cells were incubated at 1 × 10^4^ cells per well in an ultralow-attachment 6-well plate with MammoCult^TM^ culture medium (StemCell Technologies, Vancouver, BC, Canada), supplemented with hydrocortisone and heparin for 7 days. The cancer cells were incubated for 7 days in a 5% CO_2_ incubator at 37 °C. Mammosphere formation was quantified using the NICE program [30]. Mammosphere formation was determined by examining the mammosphere formation efficiency (MFE) (%) [31].

### 4.6. Cell Proliferation Assay

Breast cancer cells were seeded at 1.5 × 10^4^ cells per well in a 96-well plate for 24 h and incubated with betavulgarin (0, 50, 100, 200, 300, 400, or 500 µM) for 24 h. Then, proliferation was assayed using the EZ-Cytox Kit (DoGenBio, Seoul, Korea) in accordance with the manufacturer’s protocol. The optical density at 450 nm (OD_450_) was measured using a VERSA max microplate reader (Molecular Device, San Jose, CA, USA).

### 4.7. Colony Formation Assay

MDA-MB-231 and MCF-7 cells were cultured at a low density (2 × 10^3^ and 3 *×* 10^3^ cells/well) in a six-well plate and treated with betavulgarin in DMEM. After 7 days of incubation, the medium was replaced, and the cells were washed with PBS, fixed with 3.7% formaldehyde, and stained for 15 min with 0.05% crystal violet. Images were acquired using a scanner.

### 4.8. Assessment of CD44^+^/CD24^-^ Expression

We used a previously described method [32]. After incubation with betavulgarin for 24 h, MDA-MB-231 cells were harvested and dissociated. Next, 1 *×* 10^6^ cells were labeled with FITC-labeled anti-CD44 and PE-labeled anti-CD24 antibodies (BD), and incubated at 4 °C for 20 min. Then, the cells were washed two times with 1 *×* PBS and assayed using a cytometer (Accuri C6, BD, San Jose, CA, USA).

### 4.9. Transwell Assay

We followed a previously described method [33]. Migration assays were performed with 12-well hanging inserts (Merck Millipore, Darmstadt, Germany). MDA-MB-231 cells were suspended in 200 µL of DMEM containing 1% FBS and added to the upper chamber (2 × 10^5^ cells/chamber). The bottom chamber was filled with 750 μL of DMEM containing 20% FBS. The cells were incubated for 24 h at 37 °C in a 5% CO_2_ incubator. The lower surface of the inserts was fixed with 3.7% paraformaldehyde and stained with 0.03% crystal violet. Images were captured with a light microscope.

### 4.10. Real-Time RT-qPCR

We used a previously described method [34]. RNA was extracted from MDA-MB-231 cancer cells and mammospheres and purified. Real-time RT-qPCR was performed with a one-step RT-qPCR kit (Enzynomics, Daejeon, Korea). The specific primers are described in Appendix A.

### 4.11. Immunofluorescence (IF)

Breast cancer cell lines were fixed with 4% paraformaldehyde for 20 min, permeabilized with 0.5% Triton X-100 for 10 min, blocked with 3% bovine serum albumin (BSA) for 30 min, and stained with an anti-p-stat3 antibody (#9145, Cell Signaling Technology, Danvers, Massachusetts, USA), followed by a secondary anti-rabbit Alexa 488-conjugated antibody (A32723, ThermoFisher, Waltham, MA, USA). The nuclei were stained with DAPI, and pStat3 was visualized with a fluorescence microscope (Lionheart, Biotek, VT, USA).

### 4.12. Western Blotting

Proteins derived from MDA-MB-231 mammospheres treated with/without betavulgarin were separated using 10% SDS-PAGE and transferred to a PVDF membrane (Millipore, Billerica, MA, USA). Membranes were blocked in Odyssey blocking buffer in PBS-Tween 20 (0.1%, *v*/*v*) at room temperature for 1 h. The blots were incubated at 4 °C overnight in a blocking solution containing the following primary antibodies-anti-p-Stat3 (#9145, Cell Signaling Technology, Denver, CO, USA), anti-p65 (LF-MA30327), anti-stat3 (sc-482), anti-Sox2 (sc-365923), anti-Lamin B (sc-6216), and anti-β-actin (sc-47778) (Santa Cruz Biotechnology, Dallas, TX, USA). After the membranes were washed with PBS-Tween 20 (0.1%, *v*/*v*), the membranes were incubated with IRDye 800CW- and 680RD-conjugated secondary antibodies, and the band signals were detected using an ODYSSEY CLx instrument (LI-COR, Lincoln, NB, USA).

### 4.13. EMSA

Nuclear extracts were prepared as described previously [35]. An EMSA for Stat3 binding was performed using an IRDye 700-labeled Stat3 DNA (LI-COR). Samples were run on a nondenaturing 5% PAGE gel, and EMSA data were captured with an ODYSSEY CLx instrument (LI-COR).

### 4.14. Statistical Analysis

All presented data are the mean *±* standard deviation (SD). Data were analyzed using Student’s *t*-test. A *p*-value less than 0.05 was considered statistically significant (GraphPad Prism 5 software).

## 5. Conclusions

A BCSC-inhibiting compound from beet extracts was purified using silica gel, gel filtration, TLC, and HPLC. The compound was identified as Betavulgarin, a mammosphere formation inhibitor, was isolated from beetroot and identified by mass and NMR spectroscopy. Betavulgarin inhibited cell proliferation, BCSC formation, and reduced the size of the CD44^+^/CD24^−^ subpopulation and the transcript levels of the C-myc, Nanog, and Oct4 gene. This compound decreased the nuclear localization of Stat3 and reduced the mRNA and protein levels of SOX2 in mammospheres. Our results in this study showed that betavulgarin inhibited the Stat3/Sox2 signaling pathway and induced BCSC death, indicating that betavulgarin might be a potential natural compound that targets breast cancer and BCSCs.

## Figures and Tables

**Figure 1 molecules-25-02999-f001:**
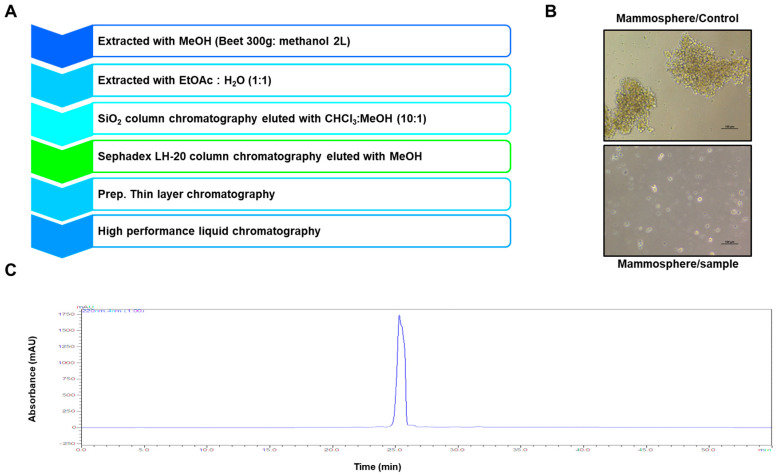
Identification of a breast cancer stem cell (BCSC) inhibitor isolated from beet extracts via a mammosphere formation assay. (**A**) Isolation procedure for the mammosphere formation inhibitor. (**B**) Assay for mammosphere formation inhibition using beet extracts. Mammospheres were incubated with beet extracts or DMSO. MDA-MB-231 cells were treated with beet extracts or DMSO in BCSC culture medium for seven days. Images were obtained by microscopy at 10× magnification and show representative mammospheres (scale bar=100 µm). (**C**) HPLC chromatogram of the inhibitor isolated from beet extracts.

**Figure 2 molecules-25-02999-f002:**
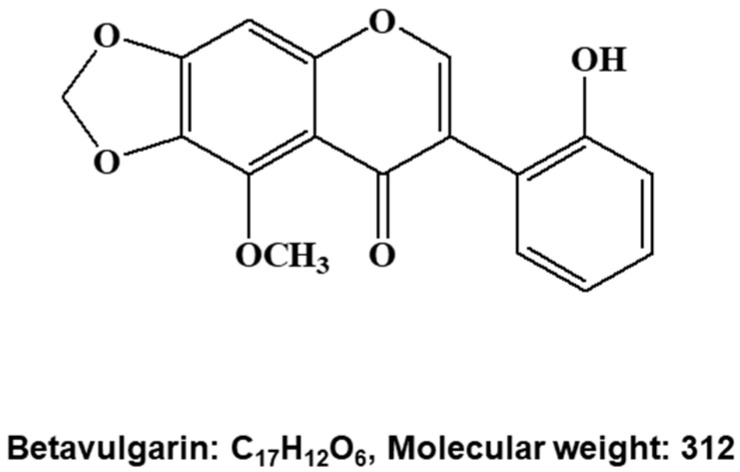
Molecular structure of the BCSC inhibitor isolated from beet extracts. Molecular structure of betavulgarin.

**Figure 3 molecules-25-02999-f003:**
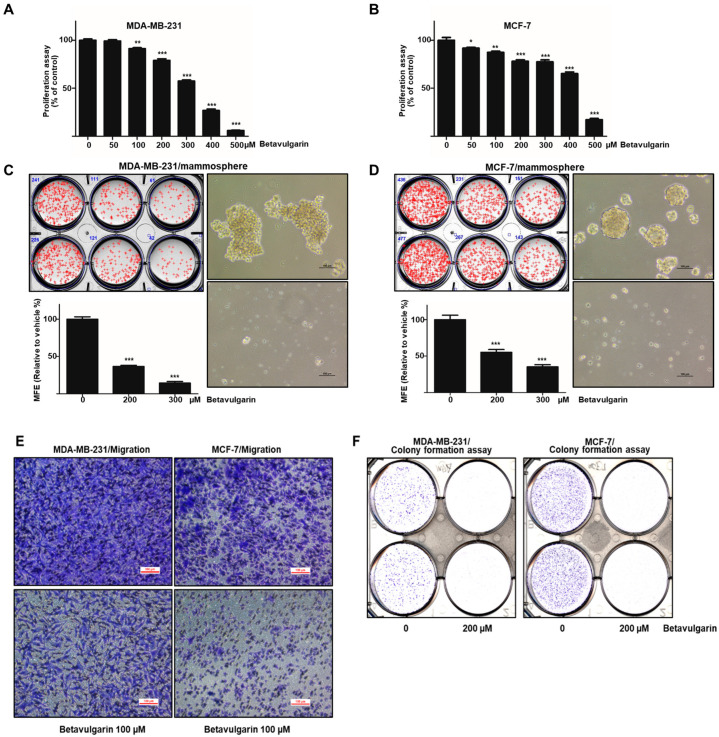
Effects of betavulgarin on cancer cell growth and mammosphere formation. (**A**) MDA-MB-231 cells were treated with betavulgarin in culture medium for 24 h. A cell growth assay using betavulgarin was performed with an EZ-Cytox kit. (**B**) Breast cancer MCF-7 cells were treated with various concentrations of betavulgarin in a culture medium for 24 h. The cell proliferation of the MCF-7 cells was measured with an EZ-Cytox kit. (**C** and **D**) Betavulgarin inhibits the formation of mammospheres. For the establishment of mammospheres, 1 × 10^4^ MDA-MB-231 cells or 4 × 10^4^ MCF-7 cells were seeded in ultralow-attachment 6-well plates in BCSC culture medium. The mammospheres were incubated with 200 µM or 300 µM betavulgarin or DMSO for seven days. Representative images of mammospheres were obtained by microscopy (scale bar: 100 µm). The mammosphere formation efficiency (MFE) was examined. (**E**) Transwell assays were performed to determine the cell migration of MDA-MB-231 and MCF-7 cells exposed to betavulgarin (scale bar: 100 μm). (**F**) Betavulgarin inhibits colony formation by MDA-MB-231 and MCF-7 cells. The cancer cells were incubated in 6-well plates and treated with betavulgarin for 7 days. Representative data were collected. The data from triplicate experiments are represented as the mean ± SD; * *p* < 0.05; ** *p* < 0.01; *** *p* < 0.001.

**Figure 4 molecules-25-02999-f004:**
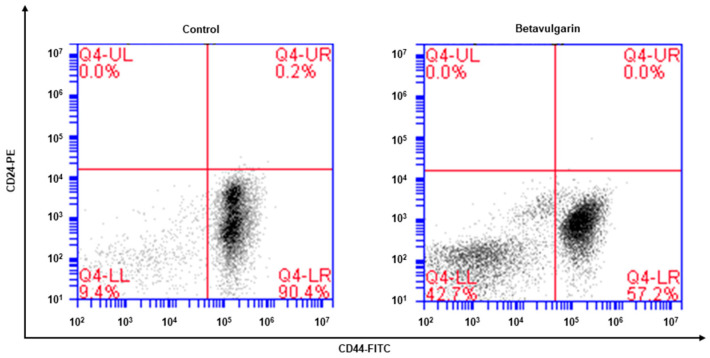
Effect of betavulgarin on the proportion of CD44^high^/CD24^low^ cells. The CD44^high^/CD24^low^ cell population of MDA-MB-231 cells treated with betavulgarin (200 μM) or DMSO for 24 h was assessed. A total of 1 × 10^6^ cells were analyzed by flow cytometry. The gating was based on the binding of a control antibody (Red Cross).

**Figure 5 molecules-25-02999-f005:**
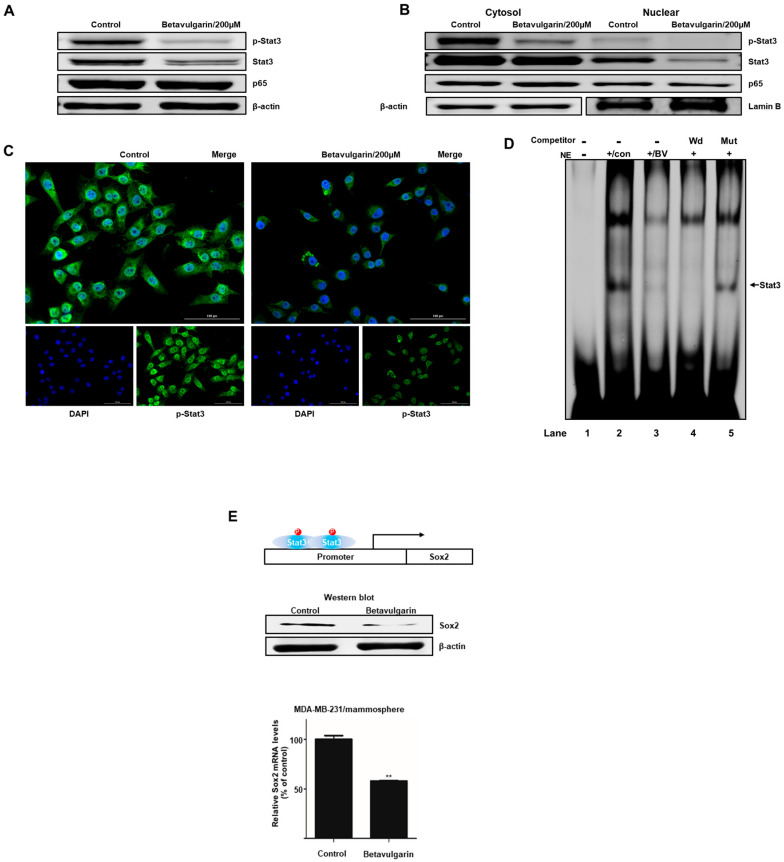
The effect of betavulgarin on the Stat3 signaling pathway. (**A**) The levels of p-Stat3, Stat3, and p65 in isolated total protein from MDA-MB-231-derived mammospheres were measured after treatment with betavulgarin for 48 h, using Western blot analysis. (**B**) The nuclear protein levels of Stat3 and NF-κB were determined in MDA-MB-231-derived mammospheres treated with betavulgarin (200 µM) or DMSO. Betavulgarin blocked the translocation of Stat3 and decreased the level of p-Stat3 in mammospheres. (**C**) Immunofluorescence (IF) analysis of p-Stat3 (green) expression and localization in breast cancer cells treated with betavulgarin or DMSO was performed (scale bar: 100 µm). (**D**) Electrophoresis mobility shift assays (EMSAs) of MDA-MB-231-derived mammosphere nuclear proteins after treatment with betavulgarin were performed. Nuclear extracts were incubated with a Stat3 probe and separated by 5% native PAGE. Lane 1-Stat3 probe only; Lane 2-untreated nuclear extracts with the Stat3 probe; Lane 3-betavulgarin-treated nuclear proteins with the Stat3 probe; Lane 4-untreated nuclear proteins incubated with a self-competitor (100×) oligo; and Lane 5-untreated nuclear extracts incubated with a mutated-Stat3 (100×) probe. The arrow indicates the DNA/stat3 complex in the mammosphere nuclear lysates. (**E**) Treatment of tumorspheres with betavulgarin (200 µM) for 48 h decreased the mRNA and protein levels of SOX2. The transcription of the SOX2 gene was assayed with specific real-time RT-qPCR primers. The SOX2 protein was identified with an anti-SOX2 antibody. β-actin was used as an internal control. The data are presented as the mean ± SD of three independent experiments. ** *p* < 0.05 versus the DMSO-treated control group.

**Figure 6 molecules-25-02999-f006:**
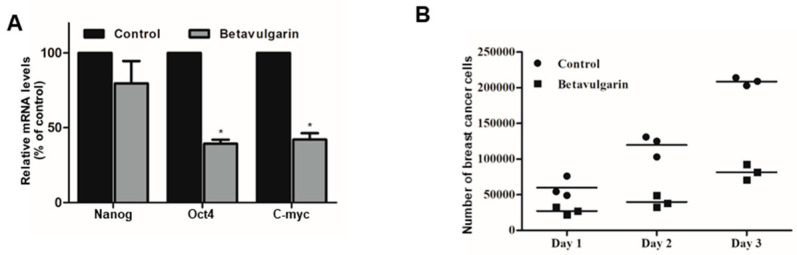
Effects of betavulgarin on the expression of BCSC marker genes and mammosphere growth. (**A**) Real-time qPCR analysis of the Nanog, c-Myc, and Oct4 genes in mammospheres, after treatment with betavulgarin for 46 h. (**B**) Mammosphere growth was inhibited by betavulgarin. Mammospheres with/without betavulgarin were divided into single cells and plated in 6-well plates in equal numbers. One, days 2 and 3, the cells were counted. The data from triplicate experiments are represented as the mean ± SD. * *p* < 0.05 versus the DMSO-treated control group.

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
