# Peer review of "Betavulgarin Isolated from Sugar Beet (Beta vulgaris) Suppresses Breast Cancer Stem Cells through Stat3 Signaling"

_molecules, 2020, doi:10.3390/molecules25132999_

Round 1

Reviewer 1 Report

Authors describes the effect of betavulgarin on breast cancer stem cells. The applied methodology is correct however some major revision re required before publication.

  1. The abstract is too descriptive. Please insert the best obtained data and the novelty of this work.
  2. Please modify in Keywords “cancer stem cells (CSCs)” with “breast cancer stem cells (BCSCs)”
  3. Improve discussion section. Actually is too general and without any link with the results that are obtained. Are other natural products able to exert a similar activity? Which is the differences with betavulgarin?
  4. Modify paragraph 4.1. Chemical and Reagents by remove each apparatus that should be detailed were was applied. Example remove: HPLC was performed with a Shimadzu application system (Shimadzu, Kyoto, Japan).
  5. Explain better “urban farmers”

Modify the title “4.3. Isolation of an Inhibitor” as well as 4.4 an inhibitor is generic

  1. Indicates MCF-7 and MDA-MB-231 breast cancer cell lines were purchased from the American Type

Culture Collection numbers in bracket after each cell line.

  1. Conclusion are lacking of content. Please rewrite it in order to highlight the novelty of this study. The best obtained results and also further perspective of this knowledge.

Author Response

We attached reviewer1's comments.

Reviewer 2 Report

The manuscript by Liu et al. described a tumor-suppressive effect of Betavulgarin targeting cancer cell cells in breast cancer. The authors isolated Betavulgarin from sugar beet by several processes of chromatography. Betavulgarin inhibited mammosphere formation, migration, and colony formation as well as the reduction of cancer stem cell fraction in MDA-MB-231 cells. The treatment of Betavulgarin inhibited p-Stat3 and following sox2 expression, similar to other CSC markers such as cMYC and Oct4. The experiment was properly performed and the manuscript was well-written. Specific concerns are shown below.

  1. Have the authors tested Betavulgarin on normal mammary epithelial cells such as HMEC and MCF10A? It is important to know if Betavulgarin exerts less effect on normal cells.
  2. Figures 3E and F were not shown in the text of the result. Please describe them properly.
  3. In figure pictures, scale bars were invisible. Please revise them.

Author Response

We attached Reviewer 2 comments.

Round 2

Reviewer 1 Report

Authors modify manuscript according  to indications.  Manuscript is now suitable for publication.